# Bacterial Interactions in the Context of Chronic Wound Biofilm: A Review

**DOI:** 10.3390/microorganisms10081500

**Published:** 2022-07-25

**Authors:** Benjamin A. R. N. Durand, Cassandra Pouget, Chloé Magnan, Virginie Molle, Jean-Philippe Lavigne, Catherine Dunyach-Remy

**Affiliations:** 1Bacterial Virulence and Chronic Infections, UMR 1047, Université Montpellier, INSERM, Service de Microbiologie et Hygiène Hospitalière, CHU Nîmes, 30908 Nîmes, France; durand.b.pro@outlook.fr (B.A.R.N.D.); cassandra.pouget@chu-nimes.fr (C.P.); chloe.magnan@chu-nimes.fr (C.M.); jean.philippe.lavigne@chu-nimes.fr (J.-P.L.); 2Laboratory of Pathogen Host Interactions, Université de Montpellier, CNRS, UMR 5235, 34000 Montpellier, France; virginie.molle@umontpellier.fr

**Keywords:** biofilm, chronic wounds, colonization, cutaneous microbiota, microbial crosstalk, microbial interactions, virulence

## Abstract

Chronic wounds, defined by their resistance to care after four weeks, are a major concern, affecting millions of patients every year. They can be divided into three types of lesions: diabetic foot ulcers (DFU), pressure ulcers (PU), and venous/arterial ulcers. Once established, the classical treatment for chronic wounds includes tissue debridement at regular intervals to decrease biofilm mass constituted by microorganisms physiologically colonizing the wound. This particular niche hosts a dynamic bacterial population constituting the bed of interaction between the various microorganisms. The temporal reshuffle of biofilm relies on an organized architecture. Microbial community turnover is mainly associated with debridement (allowing transitioning from one major representant to another), but also with microbial competition and/or collaboration within wounds. This complex network of species and interactions has the potential, through diversity in antagonist and/or synergistic crosstalk, to accelerate, delay, or worsen wound healing. Understanding these interactions between microorganisms encountered in this clinical situation is essential to improve the management of chronic wounds.

## 1. Introduction

Chronic wounds are cutaneous lesions failing to proceed through the normal phases of healing in an orderly and timely manner (more than four weeks). They affect millions of patients annually. The three types of chronic wounds are diabetic foot ulcers (DFUs), pressure ulcers (PUs), and venous and/or arterial ulcers [1]. These chronic wounds share risk factors delaying healing, such as arteriopathy for vascular ulcers [2], neuropathy, excess moisture, and arteriopathy for PUs and DFUs [3].

Their management in care facilities is particularly long [4,5], placing patients at high risk for recurrent episodes [6] and longer hospital stays [7]. For example, Yao et al. observed 13 average days of hospital duration with high readmittance (10 times average) in a retrospective epidemiological study of chronic wounds [8]. In DFUs, readmissions corresponded within an average of 18.27 days (+/−21.01), representing a cumulative 228 days (~32 weeks) in hospital facilities [8]. This was in accordance with another study that found a median hospital duration of 17 weeks (interquartile range: 7–34) [9]. Moreover, these chronic wounds contribute significantly to patient morbidity and mortality [1].

The complications associated with chronic wounds exacerbate the economic burden of these wounds and reduce patients’ quality of life [10]. Among the common complications, bacterial infections are the most frequent cause of delayed healing. They can cause localized infections through systemic infections and limb-threatening infections [1]. Their management remains problematic due to the lack of diagnostic tools allowing the distinction between colonization and infection of the wounds. Microorganism culture from these wounds cannot assess infection status, as skin is colonized by commensal microorganisms [11]. Moreover, the transition from the colonized wound toward infected status is associated with the clinical manifestations of inflammation for which process might be impaired, notably in the context of DFUs in which peripherical immunopathy decreases host defense abilities [12,13,14]. Importantly, most bacteria present in chronic wounds are in biofilm status [15]. Biofilm is usually defined as a community of microorganisms attached at an interface and encased within a polymeric matrix [16]. This organization contributes to keeping the ulcer in a prolonged inflammation and non-healing state. The medical consequence of this chronic wound is the increased use of antibiotics, in turn leading to increased bacterial resistance [17,18,19]. Antimicrobial resistance is due to (i) antibiotic misuse [20,21], (ii) hospital-acquired strains exhibiting wider antimicrobial resistance [22], and (iii) the presence of biofilm at the wound level [23]. The biofilm has increased tolerance [24], selecting resistant bacteria both through pressure selection [25] and genetic transfer [26], making it a potential “powder keg” reservoir. Paradoxically, in a mice model, an off-target effect of antibiotic (affecting cutaneous microbiota in addition to the targeted pathogen) contributed to delayed healing [27].

The aim of this review was to explore the impact of microbial interactions on the evolution of an ulcer and to discuss whether the endogenous interactions underlying the biofilm could be exploited to improve clinical outcomes.

## 2. Ecology of the Chronic Wounds: An Interactome Bed

### 2.1. Ecology of Chronic Wounds

Normal skin microbiota varies greatly at inter- and intrapersonal levels but is relatively temporally stable. The main phyla detected are *Actinobacteria* (corrig. phyl. *Actinomycetota*), followed by *Firmicutes* (corrig. phyl. *Bacillota*), *Proteobacteria* (corrig. phyl. *Pseudomonadota*), and *Bacteroidetes* (corrig. phyl. *Bacteroidota*) [28,29]. Foot-skin microbiota has the least temporal stability [29] and has been classified as a moist niche enriched in *Actinobacteria* (*Corynebacteriaceae*) and *Firmicutes* (*Staphylococcaceae*) [30].

In 2000, Schmidt et al. found no specific differences between microbiota composition of diabetic lower limb ulcers (ischemic or not) and vascular ulcers and their evolution, as later corroborated by Wolcott et al. [31,32]. However, more recent studies contradict these findings. Pooling an equal number of chronic wounds (e.g., DFUs, PUs, venous, and arterial wounds) vs. healthy skin and using swabbing before and after wound debridement, Verbanic et al. observed that the taxa diversity harbored an exclusive and significant enrichment in *Proteus*, *Enterobacter*, *Helcococcus*, and to a lower extent, *Staphylococcus aureus* in wound samples. In contrast, *Kocuria* and *Micrococcus* were the only ones enriched in healthy skin samples. For strict and facultative anaerobe abundance, a statistically significant correlation was found between facultative anaerobes and non-healing wounds [33]. Tipton et al. proposed a simplified model focusing on species with at least 10% prevalence across their study. Reducing individual diversity to the 11 selected species highlighted that *Pseudomonas aeruginosa* was positively correlated with increased healing time when its abundance increased, unlike *Anaerococcus vaginalis*, which was correlated with faster healing [34]. Conversely, the genus *Anaerococcus* was linked with a worsening evolution of the wounds in other studies in which it was associated with other bacteria in the same cluster [35,36]. Moreover, *Corynebacterium* sp. could be considered a potential driver of impaired healing. Their presence was predictive of worse clinical outcomes [37], and the use of targeted antibiotic treatment (e.g., clarithromycin) reduced the healing time [38]. On the other hand, wounds colonized with *Staphylococcus epidermidis* exhibited higher diversity than those colonized by *P. aeruginosa*, which showed the lowest diversity [34].

In contrast, Dowd et al. surveyed the total microbial diversity recovered from debridement material of DFUs, venous ulcers, and PUs. This material comprises devitalized tissues and biofilm-encased microorganisms. To account for interindividual diversity, they pooled extracted DNAs of each 10-patient sample by wound type and illustrated abundance according to respiratory type. Pyrosequencing showed that strict anaerobes were most prevalent in PUs, accounting for 62% of the diversity, contrary to venous ulcers in which they accounted for less than 1.6% (30% for DFUs). The DFU niche contained the fewest strict aerobes, with around 7.8% [39].

More generally, and focusing on DFUs, compared with healthy skin, microbiota showed an average increase in *Firmicutes* and *Fusobacteria* to the detriment of *Bacteroidetes* and *Proteobacteria* [35]. *Peptoniphilus*, *Staphylococcus*, *Anaerococcus*, and *Corynebacterium* were observed at higher frequencies in new ulcers and, except for *Staphylococcus*, from recurrent ones. Most of the common bacteria genera identified in new and recurrent wounds were Gram-positive and mostly cocci. The most frequent Gram-negative species detected was *Porphyromonas* spp., DFUs’ most frequently harbored facultative and strict anaerobes. Strict anaerobes had a higher representation in recurrent ulcers [40]. Gardiner et al. found that the difference between normal and diabetic foot microbiota was marked by a decrease in diversity but similar abundancy compared with a healthy population. Furthermore, they highlighted the similarity between the cutaneous microbiota of the wound and that of the undamaged skin and, strikingly, a concordance between the most represented OTUs, with *Staphylococcus* sp. as the leading representant [41]. However, different parameters could influence the wound microbiota. Jnana et al. reported that poor glucose control (HbA1c > 7.5%) was associated with a higher representation in *Actinobacteria* (9.6%) [42]. In their study centered on non-healing wounds, Choi et al. observed an overall taxonomic assignation similar to previous studies. They also explored associations between obligate anaerobes forming a cluster. *Anaerococcus*, *Finegoldia*, *Peptoniphilus*, *Porphyromonas*, and *Prevotella* had a strong Pearson correlation score, illustrating their linear correlation, and occasionally formed the major or only species recovered from some samples [43]. Interestingly, a higher abundance of *Peptoniphilus* significantly correlated with impaired wound healing, with a similar trend, although not reaching significance for *Finegoldia* and *Anaerococcus* [35]. These results were corroborated by a recent study focusing on PUs, underlying the detrimental presence of *Peptoniphilus* but also *Proteus*, *Morganella*, and *Anaerococcus* as a worsening cluster [36].

Notably, among all the microorganisms identified by the microbiota studies, *S. aureus* remains the most prevalent pathogen [44]. In a meta-transcriptomic analysis of DFUs at various infectious stages, this bacterium represented one of the main virulence factor producers [45].

Finally, chronic wounds demonstrate diversity loss in the skin microbiota, with a median of five species [32], associated with a dominance effect [41] of one (or two) species per wound. However, no particular microbiota pattern seems to be associated with infection outcomes [46]. These different species organize themselves in the wound bed in a polymicrobial biofilm [23,47,48] (Figure 1). The composition of this reservoir participates in the chronicity of the lesions, thwarting the host immune response [49] and antibiotic treatment [50]. Moreover, specific associations of commensal and pathogenic bacteria seem to induce worse prognoses for wounds; these are referred to as functionally equivalent pathogroups [36,48].

### 2.2. Factors Influencing the Dynamics of Chronic Wound Microbiota

Wound management includes debridement and, if needed, antibiotic therapy. In 2017, Tipton et al. followed community evolution in a pooled subset of chronic wounds (DFUs, PUs, venous ulcers, and surgical wounds). They included 167 subjects who underwent three hospital consultations (with an average of 149 intervisit days,) associated with wound debridement for each visit. This allowed a “long-term” view on the role of debridement on microbiota disruption [51] compared with Verbanic et al. (2020), who highlighted a marginal effect on the microbiota immediately after debridement [33]. Tipton et al. observed a median of 19 bacterial species in different samples, with the major representatives being *S. aureus*, *P. aeruginosa*, *Staphylococcus haemolyticus*, *Corynebacterium striatum*, *S. epidermidis*, and *Finegoldia magna*. During the study, and at the individual level, a depletion effect of the most abundant species of the microbiota was noted, with more than 50% of the species differing since the previous sampling time point. This transition was marked by a dominance effect (>10% abundance of the sample) of previously under-represented species, found at low abundance (<1%), in 16% to 20% of the later samples. The results also revealed that 47% of the wounds were constantly transitioning, with low-abundance species becoming more common. In addition, 80% of the patients showed a transition from low-abundance species toward common or dominant ones [51].

In 100 DFUs studied by Loesche et al., *Staphylococcus* spp. (respectively *S. aureus* and *S. pettenkoferi*) was the most abundant genus, followed by *Streptococcus*, *Corynebacterium*, and *Anaerococcus*. The authors reported a community organization in which intercommunity type (CT) differences relied on the top five abundant taxa. The major differentiating taxa were *Staphylococcus* and *Streptococcus*. They defined four CTs: two heterogeneous presenting no dominance (both enriched in *Staphylococcus* and *Corynebacterium*, CT1 slightly more and at the expense of *Anaerococcus* compared with CT2), whereas CT3 was dominated by *Streptococcus*, and CT4 showed increased abundance in *Staphylococcus* (specifically *S. aureus*). 

Patients were clustered into groups according to clinical outcome: Those having undergone amputation were located in the CT1 or CT2 group, and those with unhealed wounds were in the CT4 group. The more often patients transitioned from one CT to another, the faster they healed. This was correlated with visit frequency, which included debridement and further dressing of the wound. However, they did not find any microbiota variations (in diversity or richness) associated with antibiotic administration. Nevertheless, the group that received antibiotics to manage DFUs presented a greater community disruption compared with the group that received antibiotics to manage other infections (i.e., urinary tract infections). In addition, DFUs presented extensive modifications of their microbiota, and antibiotics provided an additive effect potentiating the community disruption of the wound microbiota [52].

In their 26-week follow-up study, Kalan et al. described *Staphylococcus*, *Corynebacterium*, *Pseudomonas*, and *Streptococcus* as the most prevalent genera. Debridement acted mostly on low abundant taxa, marginally decreasing anaerobe abundance, as diversity observed during pre- and post-treatment remained similar in non-healing wounds. Contrary to antibiotics, debridement lowered community relative abundance, leading to wound improvement [53], as described by Loesche et al. [52]. This reduction in bacterial diversity mainly concerned anaerobes. Moreover, this study highlighted a high abundance of genes involved in virulence and defense pathways, and a strong association between microaerobiosis/anaerobiosis environment and virulence. They also noted an enrichment in *S. aureus* biofilm-associated genes in non-healing wounds [53].

Finally, debridement appears to present a delayed action, as microbiota detected before and immediately after debridement did not show major differences [33]. However, over weeks, low-abundance genera emerged to replace the originally abundant genera [51]. Furthermore, higher microbiota transition frequency (without a dominance effect) is associated with better wound prognosis. This transition is associated with a shift in preferential community organization type (CT) [52]. Moreover, the environment influences the evolution of microbiota, with the example of deep and poorly oxygen wounds showing increased virulence genes [53]. In addition, the presence of anaerobes was related to ulcer depth [52] and to poor prognosis [35]. Finally, the spatial organization of the coexisting species in wounds was influenced by nutrient availability [54,55], compounds such as dioxygen (anoxic at the wound center or within the biofilm [56] and hypoxic at the edge [57,58]), and biofilm structures [59,60] (Figure 1). The inter-relationships between the different species present in the wound affect their microbiogeography either by competing [61], repelling [62], cheating [63,64], and/or attracting [65] each other. Different techniques may be used to topographically explore the organization of the various species found in these wounds such as direct visualization assisted with microscopy (e.g., PNA-FISH [23,66], mixed-culture model of genetically engineered strains), or metagenomics applied at various spots of the wound [67,68].

### 2.3. Kinetics of Wound Bed Colonization

*S. aureus*, *P. aeruginosa*, *Enterococcus faecalis*, and *Finegoldia magna* were used for polymicrobial biofilm assay via an in vitro constituted biofilm transplanted on a mouse wound bed. *P. aeruginosa* colonized the wound alone at its margin and throughout the wound surface, while the other bacteria created deep hotspots of colonization with mixed communities at their periphery. In this model, the representation varied over time, starting with an early over-representation of *P. aeruginosa*, then replaced (increased relative abundancy) by *F. magna* and *S. aureus* by the endpoint (8 and 12 days) [69]. *P. aeruginosa* hijacked the early aggregation of *S. aureus* to aggregate itself, and *S. aureus* showed a higher capacity to adhere to host-immobilized protein than *P. aeruginosa.* Initially, *S. aureus* directly disrupted an immature biofilm of *P. aeruginosa* but, once established, favored the secondary attachment of *P. aeruginosa* [65]. An in silico prediction of community composition based on metabolic modeling predicted *S. aureus* as the main representant, and the prediction of the mutualistic association between *P. aeruginosa* and *S. aureus* (possibly enhanced by some commensal) was in favor of *P. aeruginosa*. *Acinetobacter* was predicted to be fed by *S. aureus* products (explaining its absence when *P. aeruginosa* was present) [70]. In addition, the *F. magna*–*P. aeruginosa* pair was positively correlated, whereas *Staphylococcus* sp. and *Candida* sp. were negatively associated [71].

### 2.4. Host Factors Modulating Wound Microbiota

Tipton et al. showed that age influenced the wound microbiota. Thus, older patients had a significant presence of *S. epidermidis*, *C. tuberculostearicum*, and *S. agalactiae* [34]. In addition, glycemic control and the duration of diabetes affected the dominance and diversity of the microbiota. Lower levels of HbA1c and recent diabetes were associated with higher diversity of this microbiota, whereas the high HbA1c level and a long lifetime with diabetes increased the predominance of some genera in the wound microbiota [40] with a higher abundancy in *Actinobacteria* [41] or *Streptococcus* [52].

In terms of the host immune system, culture of primary keratinocytes in cell-free supernatant isolated from a biofilm of *Alcaligenes faecalis* increased the production of growth factors (granulocyte colony-stimulating factor (G-CSF), granulocyte monocyte colony-stimulating factor (GM-CSF), and transforming growth factor (TGF-α)) and proinflammatory cytokines (interleukin 6 (IL-6), tumor necrosis factor α (TNF-α), and interferon-gamma-induced protein (IP-10)). Transposed in a mouse model, a monospecies biofilm of this bacterium accelerated wound recovery at day 7 (normal wound healing process), demonstrating a potential protective role of *A. faecalis* [53]. In addition, albumin, recently described to play a role in innate immunity [72], could promote the bacterial coexistence of *P. aeruginosa* and *S. aureus* [73,74].

Finally, we should note that *S. aureus* was the most prevalent bacteria isolated in chronic wounds, notably in DFUs. The nasal carriage of this bacterium is an important niche and could represent the origin of this high colonization [75].

### 2.5. Role of Mycobiome in Chronic Wounds

The wound surface area shelters numerous microorganisms, including fungi. A recent study in Ghana found that 22% of the wounds were positive for the presence of fungi with *Candida* spp. as the most frequently observed [76]. Dowd et al. have previously reported that *Candida* were the most abundant fungi detected in chronic wounds, with the presence of *C. albicans* and *C. parapsilosis* as top species. Other species have been observed such as *Malassezia restricta* and *Curvularia lunata* and some fungi not previously recognized as human pathogens (e.g., *Pyrenophora* and *Myrothecium*) [71].

Kalan et al. studied the evolution of DFU microbiota in 100 patients every 2 weeks over 13 visits. Among the 482 OTUs identified, 17 belonged to *Ascomycota* (14/17) or *Basidiomycota* (13/17) phyla. Ten OTUs corresponded to environmental *Ascomycota*. In addition, this phylum contained the two most abundant species detected: *Cladosporium herbarum* and *Candida albicans* (which were found in 41% and 22% of the samples and 57% and 47% of the patients, respectively). The most abundant *Basidiomycota* were the yeasts, *Trichosporon* and *Rhodosporidium* spp. Mycobiome diversity varied with the localization of the DFU. Indeed, a significantly lower Shannon index (microbiota diversity) was observed at the hind foot compared with the forefoot; this was correlated with an increased abundance of *C. albicans* at the hind-foot location. Noticeably, the presence of yeast in microbial culture was associated with complications of the wounds, although the number of cases remained low (*n* = 5). Pathogenic fungi were most abundant in necrotic tissue and were associated with non-healing wounds, representing a biomarker of poor wound evolution. In these mixed biofilms, fungi constituted the core, and bacteria coated the yeast. *Corynebacterium* sp. were negatively correlated with *C. albicans* and *C. parapsilosis*, whereas *C. albicans* was positively associated with the *Alcaligenaceae* [77].

A recent study showed that fungi were differentially abundant between diabetic and normal cutaneous microbiota, with *Ascomycota* and *Basidiomycota* the most frequently identified. However, in the diabetic skin microbiota, fungi were found in lower abundance, with the noticeable presence of *Alternaria alternate* and *Cladosporium flabelliforme*, particularly enriched with *Trichophyton rubrum* [78]. In a burn rat model, the authors reported a higher incidence of fungi compared with the control. *Ascomycota* and *Basidiomycota* were always the most abundant phyla observed. *Candida* (24.8%) was the dominant genus in the burn group, compared with *Pleosporaceae* (11.0%) which was mostly identified in the healthy group. A network analysis of mycobiome showed higher stability in the healthy group [79].

### 2.6. Presence of a Cutaneous Phageome

To date, very few studies are available on the cutaneous virome and most are focused on healthy skin. The presence of the virus in chronic wounds is not documented; only the phageome has been reported.

In 2014, Oh et al. published a complete inventory of skin microorganisms and their location in healthy individuals. The authors described a low abundance of viral genomes (around 10% or less) in the interdigit space. Interestingly, *Staphylococcus* phages were preferentially found at the toenail location and in moist ecological niches (e.g., foot) [80]. Hannigan et al. studied the virus representation and interaction within the skin microbiota. They observed that viruses could be a major component of the human microbiota but were poorly present on the skin. They described a large representation of *Caudovirales*, notably in feet, where they accounted for around 75% of the relative abundance, and some *Staphylococcus* phages with a tropism of the foot, where they niched in a moist, sebaceous, and occluded microenvironment. At the toe location, the virome contained *Pseudomonas* and *Staphylococcus* phages and hits assignable to multiple species (corresponding to temperate phages [81]). This phageome was often associated with *Staphylococcus*, *Corynebacterium*, and to a lesser extent, *Cutibacterium* (formerly *Propionibacterium*). The dynamic comparison of phageomes showed a stable core virome shared between all parts of the body and some niches at the personal and interpersonal levels [82]. In 2017, the same authors described the interaction dynamics between phages and their host. The study focused on phage hypervariable loci present on skin virome. In total, 465 hypervariable loci of *Staphylococcus* phages were detected, mainly related to tropism, host immune evasion, and host resource manipulation. Those regions were predicted to be weakly functionally affected due to deleterious amino acid substitutions (mean = 17%) [83].

Recently, Verbanic et al. focused their study on chronic wounds and reported the importance of phageome, with 15.2% and 7.5% of the total reads mapping with viral genomes, for normal skin and chronic wounds, respectively [84]. Two *Staphylococcus* phages specific to *S. aureus* were explored: one “generalist” associated with variation in the healing time of the wounds and one “specialist” associated with unhealed outcome. Both phages were specific to strains belonging to the USA400. These strains mainly differed from each other by phage insertion, the “specialist” possessing an incomplete PT1028-like phage encoding for *sec2* and *sea* genes. The “generalist” strain had two insertions including an intact *Staphylococcus* phage 96 [53]. Phage ecology at the wound level can greatly influence the virulence of *S. aureus* in DFUs by altering iron metabolism [85,86].

## 3. Crosstalk between Species in Chronic Wounds 

### 3.1. Staphylococci

As previously noted, *S. aureus* is the main pathogen isolated from chronic wounds. One of its virulence factors is the quorum-sensing-related auto-amplified loop mediated by the *agr* system, which plays an important role in chronic wounds [87,88,89,90] (Figure 2). It is important to note that this system is quite conserved among *Staphylococcus* sp. [91]. Indeed, four groups of auto-inducers and their cognate receptors have been described [92], creating cross-inhibition/activation patterns [93]. For example, *S. epidermidis* auto-inducing peptides (AIPs) showed a more potent inhibitory activity on the *S. aureus agr* system than *S. aureus* AIPs on *S. epidermidis agr*, which can help colonize the wound niche. Notably, the subtype 4 AIP receptor appeared to be less sensitive to *S. epidermidis* antagonism [94]. This interplay of secretory proteins allows crosstalk between species and can also lead to predatory behavior; some *S. aureus* subpopulations (called “cheaters”) exploit *agr* stimulation potential in the absence of their own AIP secretion [95]. This behavior was not restricted to AIP interplay. Indeed, the proteome profile of coisolated subspecies highlighted that bacteria possessed different adaptive responses to the environmental conditions, including clear virulence potential, and others involved in bacterial homeostasis and metabolism to adapt to the conditions (persistence) and colonize the site [96].

In addition to AIP interplay in the nostrils, *S. aureus* has been shown to interact with *S. epidermidis*, through secretion of Esp serine protease. During this interaction, *S. epidermidis* inhibits and destroys the *S. aureus* biofilm [97]. The Esp protein possesses a specific proteolytic action on two main virulence factors produced by *S. aureus*, FnBPA, and Spa. These *S. aureus* proteins play important roles in cell adhesion, a preliminary step in biofilm formation. Both *S. aureus* Eap [98] and *S. aureus* autolysin [99] are targets of Esp. 

Another interesting interaction in the nostrils has been recorded between *Staphylococcus lugdunensis* and *S. aureus* upon iron starvation (Figure 2). Indeed, upon the low-to-absence availability of iron, *S. lugdunensis* secrets a colonization inhibitor, the lugdunin, blocking the potential competitor for this nutrient, potentiated by the recruitment of immune cells [62,100]. This effect is particularly interesting due to the correlation between the presence of *S. aureus* in nasal passages and DFU carriage [75,101] and the high level of *S. lugdunensis* in chronic wounds (associated with *S. aureus*) [102].

### 3.2. S. aureus and P. aeruginosa

The most extensively described interaction within the chronic wounds ecosystem is the competition/synergy between *S. aureus* and *P. aeruginosa*. Indeed, the virulence of *P. aeruginosa* can be enhanced by N-acetylglucosamine and peptidoglycan fractions from Gram-positive bacteria [69,103,104]. In parallel, through antagonist interactions, *S. aureus* is maintained in its persistence/resistance form as a small colony variant (SCV) [105,106,107] (Figure 2). This interaction alters epithelization, contributing to wound chronicity [108]. SCVs are promoted through an in vitro antagonism, resulting in a clearance of *S. aureus*; once transposed in an in vivo context, this SCV-leading interaction induces colonization promotion [109]. One of the antagonisms is driven by the *P. aeruginosa* quorum-sensing system because hydroxyquinoline N-oxide (HQNO) exposure selects the SCV form [105], decreases the *agr* system (see Section 3.1
*Staphylococci*), and promotes biofilm production [107]. Albumin has also been shown to sequester *P. aeruginosa* quorum-sensing molecules and thus prevent *S. aureus* killing [74,110]. Finally, this competition partially depleting the *S. aureus* population results in a more controlled and resistant niche, as SCVs are more resistant to antibiotics (by reducing metabolism and interspecific small molecule secretion [111,112,113], virulence factor production, and by constituting a multispecies biofilm). *P. aeruginosa* has defective virulence (e.g., protease secretion, rhamnolipids production) and biofilm function at the chronic wound level. This was highlighted through a phenotypic screening comparing specimens from wounds, acute infections, and environmental settings. From the explored collection, 22% and 43% of the wound samples were deficient in biofilm and protease production, respectively, vs. 8% and 0% from acute infections and 6% and 0% in environmental samples, respectively [54]. *P. aeruginosa* biofilm formation did not appear mandatory in its implantation in the wound. Indeed, biofilm formation involves quorum sensing and exopolysaccharide production. Deletion mutants of these related genes were not associated with a decreased wound fitness upon competition assay with parenteral strain. This observation was reinforced by transposon insertion sequencing screening failing to deplete insertional mutant in those pathways for competition in a mice model. This approach highlights that other mutants exhibited better fitness in this in vivo model. These mutants were tested for phenotypic traits and were not affected by anaerobic growth (low oxygen availability) or oxidative and membrane stressing conditions [54,65]. These studies highlighted the importance of some *P. aeruginosa* virulence genes and the ability of this strict aerobe to grow in anaerobic conditions and resist oxidative stress encountered in deep wounds. 

The multispecies biofilm formed in chronic wounds is also enhanced by the alginate production of some *P. aeruginosa* strains [114,115]. Indeed, alginate production, contributing to the mucoid phenotype of *P. aeruginosa* biofilm, not only decreases its own virulence but also promotes *S. aureus* survival (Figure 2) [116]. It has also been shown that *P. aeruginosa* isolated from chronic leg ulcers commonly present the *algD* gene [117] involved in alginate production and thus offer a protective context for *S. aureus* in a mixed biofilm. When these two species cohabit, *S. aureus* virulence is downregulated, and its metabolism is driven toward anaerobic metabolism and an increased stress response. In fact, *P. aeruginosa* decreased the *lukS* transcript levels of *S. aureus* involving a metabolic stress status of this bacterium under the governance of Sigma B. This transcription factor was shown to downregulate the two main regulators of *S. aureus* virulence, *agr* and *sar* [111,118,119,120]. Sigma B is an alternative factor against the stress response of *S. aureus*, triggered through cell-wall stiffening [59] and involved in the persistency of this bacterium [121]. Upon sigma B activation (Mg^2+^ mediated), the biofilm organization of *S. aureus* adapts with (i) the spatial dissociation of *S. aureus* and *P. aeruginosa* [23], (ii) the use of a biofilm microenvironment to promote the *S. aureus* survival (via alginate production) [116], (iii) the acquisition of cation and oxygen gradients [73,122], and (iv) the orientation of *S. aureus* toward SCV and anaerobic metabolism [111,119,121]. pH also affects this interaction. In an in vitro study on DFU samples, McArdle et al. showed an increased resistance toward ciprofloxacin for both *P. aeruginosa* and *S. aureus*. *P. aeruginosa* was more resistant under acidic conditions under which alkaline ones were more favorable for the increased resistance of *S. aureus* [123].

### 3.3. S. aureus Associated with Bacillus thuringiensis or Klebsiella oxytoca

Coculture experiments commonly showed a normal *S. aureus* growth profile. This contrasts with a bovine mastitis model, where *S. aureus* strains were affected by *B. thuringiensis* bacteriocins [124]. This association with chronic-wound-originating strains decreases its exoproteome (proteins associated with the cytosolic compartment). This attenuation in the diversity of *S. aureus* secretome was also highlighted upon coculture with *K. oxytoca* (Figure 2). *S. aureus* exoproteome decreased with the low excretion of virulence factors, often specific to culture conditions. Indeed, differential virulence factors were produced depending on mono- or coculture status [96]. It must be noted that omics studies, and more specifically exoproteome studies, are tricky, as the large dataset relies on previously characterized proteins. Unfortunately, this approach eliminates, after filtering, proteins with unknown functions and, therefore, is not able to highlight new proteins potentially functionally involved in bacterial interactions. Indeed, the virulence potential of the exoproteome can be difficult to predict, as cytoplasmic protein, released upon cell lysis, can further contribute to pathogenicity, a characteristic that could not be a priori predicted [125].

### 3.4. S. aureus and Finegoldia magna

*S. aureus* and *F. magna* may offer potential agonists to hijack the host defenses (Figure 2). Indeed, the anaerobe bacterium must reach the deepest tissue of the wound, where dioxygen pressure is the lowest. Thus, *F. magna* can use its collagenase activity of the serine-type endopeptidase SufA in partnership with virulence factors produced by *S. aureus* [126]. *S. aureus* benefits from this interaction and the action of SufA to reach the bone and cause osteomyelitis [127]. Within this context, *F. magna* could modulate the impaired immune system by decreasing the B/T cell ratio and by inhibiting the calprotectin chelation [14,126]. Another effect of SufA production is to inhibit the bactericidal activity of histones [128]. This mechanism presents functional redundancy with FnBPB produced by *S. aureus*. This protein binds plasminogen and histones in a complex. This complex can cleave histones using activated plasmin preventing its bactericidal activity [129]. The production of PVL and HlgAB induce NETosis, but also the Nuc nuclease [130] by *S. aureus* prevents biofilm infiltration by neutrophils [131,132]. These two mechanisms, potentially also present in *F. magna*, had functional redundancy to maintain bacterial load against the innate immune response and, more specifically, NETs that are involved in pathogen circumscribing and decreased virulence [131,133,134].

### 3.5. S. aureus and Corynebacterium spp.

*S. aureus* is modulated through interaction with *Corynebacterium* species toward a commensalism state affecting its behavior, virulence, and fitness (Figure 2). Using RNA-seq after cocultivation, Ramsay et al. observed that the *agr* system (the global virulence factor regulation system used by *S. aureus*) was downregulated, favoring the colonization process of the wound. This downregulation was mediated through released compounds of *Corynebacterium* not yet identified [135].

### 3.6. S. aureus and Helcococcus kunzii

In a virulence *Caenorhabditis elegans* model, some *H. kunzii*, commensal Gram-positive cocci isolated from the same DFUs as *S. aureus*, altered *S. aureus* virulence. With no impact on host-associated defense gene expression, this interaction was noted in a subspecies-dependent manner by a decrease in the *agr* system expression (Figure 2). Thus, some *H. kunzii* isolates inhibited the *S. aureus* colonizer strains, but others showed a more potent action affecting the most virulent *S. aureus* strains [136].

### 3.7. S. aureus and Candida spp.

pH is an important element in the behavior and the phenotype of *C. albicans*. At a lower pH (<6), this fungus mainly forms budding bodies, whereas at higher pH (>8), it shifts to the mycelia phenotype [137], associated with the biofilm lifestyle [138]. In chronic wounds where the pH is high [139], a mixed biofilm associating *C. albicans* and *S. aureus* can be observed. Various determinants of *C. albicans* have been described to enhance the close relationship with *S. aureus* (hyphae adhesion [140], synergistic biofilm densification [141,142]), and increased survival of *S. aureus* in the presence of antibiotics [142,143] (Figure 2). *C. albicans* can also upregulate the *agr* system of *S. aureus* when the two pathogens are near in a systemic infection model [144].

## 4. Conclusions

Chronic wounds exhibit characteristics of a polymicrobial niche where bacteria are in an organized biofilm and where microorganism interactions are particularly important in modulating the virulence of pathogens to hijack immune host defense. This organization contributes to microorganism persistence and the chronicity of the wound. Many interpersonal variabilities can be observed in the microbial community composition [32], with no clear pathogens (in opposition to Koch’s postulate) but more associated species interacting to worsen or improve the wound [145]. The complex network of interactions of the species in more or less close contact elicits cooperative or antagonist effects. 

The use of targeted antibiotics was assessed to induce a more inhibitory potential of community resilience [38,52]. However, debridement remains one of the most important parts of the management of chronic wounds [51,52]. It can have an impact on community transition by either inhibiting bacterial colonization (via the production of lugdunin [62] or ESP [98]) or depleting the environment from FEP component (e.g., bacteriocin effect [124,146]), or delaying biofilm reconstitution (ESP [97]). Debridement may offer better opportunities for drastic community shifts and favor wound healing [52]. To improve treatment and offer new therapeutics alternatives, a solution could be to modulate the pre-exiting individual behavior of bacteria by decreasing their virulence potential through compounds that have yet to be discovered and characterized (e.g., in *C. striatum* [135] or *H. kunzii* [136]). Finally, the use of selected bacterial strains or phages could be a promising approach with the aim to elicit host beneficial interactions in chronic wounds [86,136].

More knowledge and mechanistic interplay are still needed to better understand the interactome of this niche to find ways to trigger phenotypic changes to improve patient prognosis.

## Figures and Tables

**Figure 1 microorganisms-10-01500-f001:**
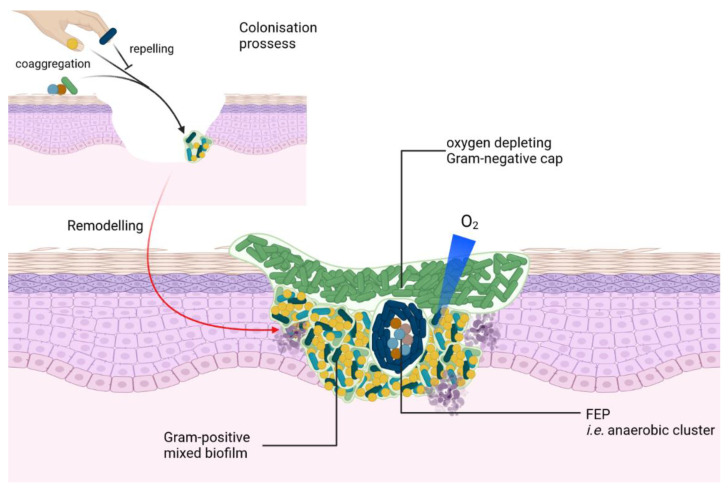
Biofilm organization at the wound level. Commensal and pathogenic bacteria colonized the wound through coaggregation/repelling either from the adjacent skin or through hand carriage. Arrows represent positive interactions, while the flathead arrow represents negative interactions. Time evolution is represented with a shaded red arrow. Bacteria are represented encased in biofilm matrix (green), which will remodel itself relying on bacterial interactions to further cover the wound. Biofilm is in contact with intact or necrotic cells of the different epidermal layers. Gram-negative, strict aerobes are represented in green and can “cap” the ulcer, and Gram-positive anaerobe/aerotolerant cocci are in yellow. Functionally equivalent pathogroups (FEPs) are illustrated at the core of *Corynebacterium* (dark blue bacilli) microniche as an anaerobic cluster of Gram-positive (brown, light brown, and blue). The dioxygen gradient is represented as a gradient blue triangle.

**Figure 2 microorganisms-10-01500-f002:**
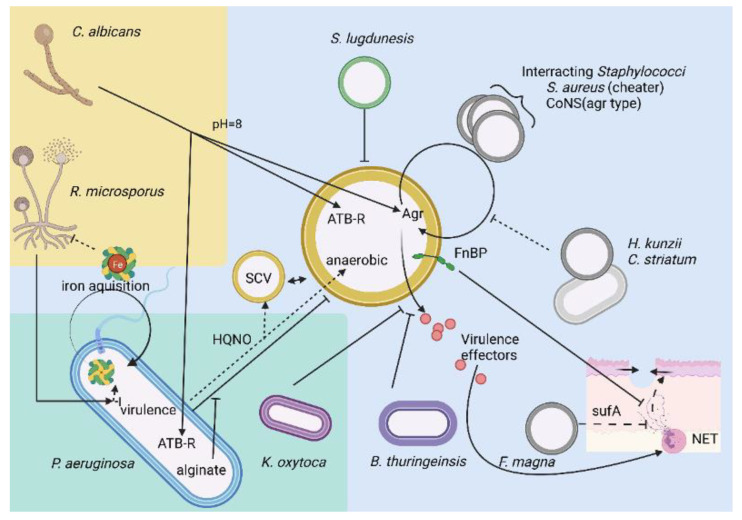
Selected network of interactions described or expected in a chronic wound environment illustrating the diversity involved within this particular niche. Fungi, Gram-negative, and Gram-positive microorganisms are highlighted in yellow, green, and blue, respectively. *S. aureus* is represented at the center of this network of interaction in yellow. Grey Gram-positive cocci represent *Staphylococci* (*S. aureus* and Coagulase negative *Staphylococci* (CoNS)). They interact through Agr system by benefiting from *S. aureus* autoinducers production and/or production of their own autoinducers antagonizing *S. aureus* Agr loop. Green/yellow helix represents siderophore used for extracellular iron acquisition. It is represented as complexed with Fe^2+^ (red core). Solid lines illustrate direct interaction between microorganisms, whereas dashed lines illustrate indirect interaction. Shaded arrows indicate dynamic interaction. Arrows highlight positive interaction, and flathead arrows highlight negative microbial interactions. ATB-R, antibiotic resistance; HQNO, hydroxyquinoline N-oxide (mediator of *P. aeruginosa* quorum sensing); NET, neutrophils extracellular trap (neutrophils are activated to enter this pathway through *S. aureus* virulence factors; after releasing bacterial DNA and histone in the extracellular environment, SufA and FnBP can degrade histones and thus inhibit their bactericidal activity).

## Data Availability

Not applicable.

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
