# Peer review of "Bacterial Interactions in the Context of Chronic Wound Biofilm: A Review"

_microorganisms, 2022, doi:10.3390/microorganisms10081500_

Round 1
Reviewer 1 Report
This review is comprehensive and well written. Although the English is sufficient, I would recommend a proofreading, to polish a few remaining inaccuracies.
Line 65: I suggest using the term microbiota instead of the outdated term microflora
Line 71: In chapter ‘Ecology of chronic wounds’ Corynebacterium should be mentioned as potential driver of impaired haling. It was shown in (Mahnic et al 2021, Front Med) that Corynebacterium was predictive of worse clinical outcome, while (Walcott et al., 2010 Immun Med Microbiol) have shown that targeted antibiotic treatment of Corynebacterium reduced the healing time.
Lines 73 and 74: Names of bacterial phyla were recently updated (https://ncbiinsights.ncbi.nlm.nih.gov/2021/12/10/ncbi-taxonomy-prokaryote-phyla-added/). New names should be at least mentioned in brackets.
Line 125: ‘poor wound evolution’ should be changed to something more concrete, maybe ‘impaired wound healing’.
Line 131: ‘this bacterium presented a significant increased expression’ needs English improvement
Line 292: In the chapter ‘Presence of a cutaneous virome’ you discuss solely phages. Is there any information on eukaryotic viruses in chronic wounds? If not, this should be mentioned. When discussing only phages you can even consider using the them phageome instead of virome.
Reviewer 2 Report
The authors in the review “Bacterial Interactions in the Context of Chronic Wound Biofilm: A Review” describe the role of biofilm and the bacterial attachment with microbial competition and/or collaboration within the chronic wounds. The review is well written and organized. It should be published with minor revisions listed below:
Line 25 “… to improve the management of chronic wounds…” the authors should argue this sentence in the manuscript
Line 48-49 the sentence is less clear please rewrite
Line 332 the acronym should be posed between the brackets “auto inducing peptides (AIP)”
Line 334 remove “ ’ ”
Line 339 the sentence seems redundant and less clear please rewrite
Line 374 the sentence should be clarified
Line 377 Please rewrite very confusing sentence
Line 386 should be augmented this sentence?
Line 388 the genes downregulated of S. aureus should be indicated
The figures should be formatted as the guidelines of the journal
Reviewer 3 Report
The topic of the present narrative review, discussing Bacterial interactions in the context of chronic wound biofilm, seems interesting and clinically relevant.
The issue has been adequately described and discussed.
The narrative review presented appears comprehensive and well structured.
Reviewed findings currently presented may pave the way for further clinical investigations and may be clinically relevant in the future in the perspective of biofilm management, therefore, I would suggest to add to conclusions section potentially effective chemical compounds.
